# DIVING SEGMENTATION MODEL INTO PIXELS

**Chen Gan**[2,3]**, Zihao Yin**[2,3]**, Kelei He**[1,3*]**, Yang Gao**[2,3]**, Junfeng Zhang**[1,3]

1 Medical School of Nanjing University
2 State Key Laboratory for Novel Software Technology, Nanjing University
3 National Institute of Healthcare Data Science, Nanjing University
{chengan,zihao.yin}@smail.nju.edu.cn, {hkl,gaoy,jfzhang}@nju.edu.cn

## ABSTRACT

More distinguishable and consistent pixel features for each category will benefit the semantic segmentation under various settings. Existing efforts to mine better pixel-level features attempt to explicitly model the categorical distribution, which fails to achieve optimal due to the significant pixel feature variance. Moreover, prior research endeavors have scarcely delved into the thorough analysis and meticulous handling of pixel-level variance, leaving semantic segmentation at a coarse granularity. In this work, we analyze the causes of pixel-level variance and introduce the concept of **pixel learning** to concentrate on the tailored learning process of pixels to handle the pixel-level variance, enhancing the per-pixel recognition capability of segmentation models. Under the context of the pixel learning scheme, each image is viewed as a distribution of pixels, and pixel learning aims to pursue consistent pixel representation inside an image, continuously align pixels from different images (distributions), and eventually achieve consistent pixel representation for each category, even cross-domains. We proposed a pure pixel-level learning framework, namely PiXL, which consists of a pixel partition module to divide pixels into sub-domains, a prototype generation, a selection module to prepare targets for subsequent alignment, and a pixel alignment module to guarantee pixel feature consistency intra-/inter-images, and inter-domains. Extensive evaluations of multiple learning paradigms, including unsupervised domain adaptation and semi-/fully-supervised segmentation, show that PiXL outperforms state-of-the-art performances, especially when annotated images are scarce. Visualization of the embedding space further demonstrates that pixel learning attains a superior representation of pixel features. The code is available here.

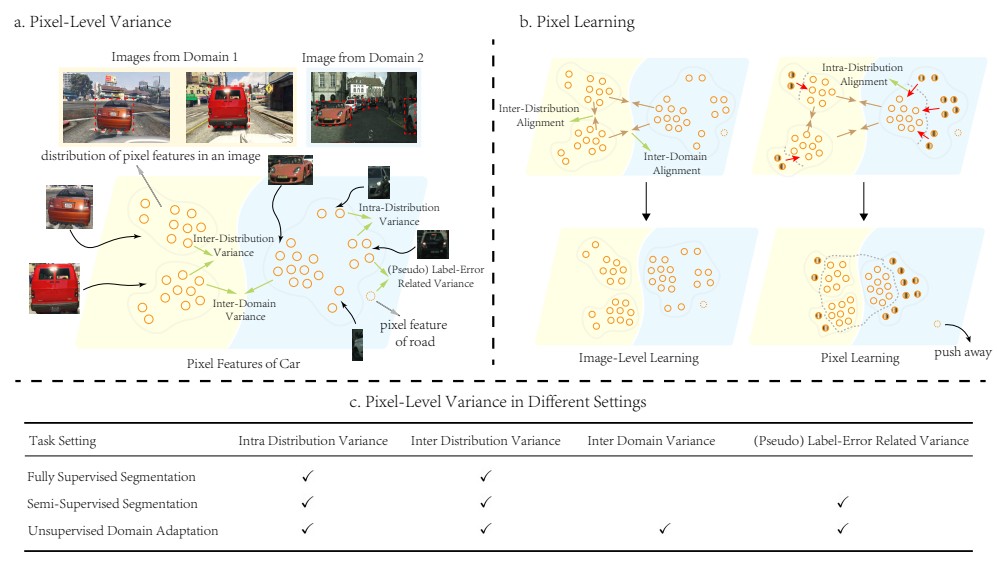

| Task Setting | Intra Distribution Variance | Inter Distribution Variance | Inter Domain Variance | (Pseudo) Label-Error Related Variance |
|---|---|---|---|---|
| Fully Supervised Segmentation | ✓ | ✓ | | |
| Semi-Supervised Segmentation | ✓ | ✓ | | ✓ |
| Unsupervised Domain Adaptation | ✓ | ✓ | ✓ | ✓ |

Figure 1: **a.** Causes of pixel-level variance. **b.** Highlights of pixel learning. **c.** Summary of pixel-level variance in different settings.

---

*: Corresponding author

# 1 INTRODUCTION

Semantic segmentation is challenging as it requires categorical pixel-level annotations precisely and consistently under different background knowledge contexts, *e.g.*, organs or lesions in medical image analysis (He et al., 2022; Liu et al., 2022), and city scenes in autonomous driving (Tsai et al., 2018; Hoyer et al., 2022a). As a dense prediction task, the pixel features within the same category across different images are expected to conform to an implicit **global distribution**. Thus, the pixel features extracted from one or several images reveal partial of the global distribution, which can be denoted as **local distribution**, and exhibits bias from global distribution. Existing research mainly focused on directly modeling or representing the global distribution using prototypes (Zhang et al., 2021; Liu et al., 2020; Yang et al., 2020) or speculated prior (*e.g.*, the Gaussian distribution (Xie et al., 2023)). However, the complexity of the global distribution exceeds the representation capacity of prototypes or primary prior distribution. The notable pixel-level variance caused by the factors summarized in Fig. 1 also aggravates the problem. The exquisite design of multiple prototypes or sophisticated prior distribution for specific scenarios also makes them infeasible and biased. We observe that successful segmentation models can construct well global distribution for categorical features during training, but pixel features within local distributions may deviate. Therefore, per-pixel alignment on deviated features helps mitigate pixel-level variance and approach global consistency.

In this work, we raise the concept of **pixel learning**, wherein each image is regarded as a distribution of pixels. Obviously, pixel features are varied in the same distribution (image), among different distributions, and different domains. The unreliable pseudo labels in unsupervised domain adaptation or semi-supervised settings also exacerbate the variance, as summarized in Fig. 1. Therefore, we propose the pixel learning scheme to address the intra- and inter-distribution (image) variance and achieve a consistent global distribution for each category eventually. Following pixel learning, we introduce a novel segmentation framework PiXL, built on a pure pixel-based learning strategy. Specifically, pixel features from several images form the local distribution at each learning step. For each class, only a subset of pixel features within the local distribution conforms to the global distribution, while the others do not, which are denoted as joint pixels and drift pixels, respectively. We first propose the Pixel-Level Sub-Domain Partition (PSP) module to discern those pixels based on their entropy. The multi-resolution prototypes are derived from the joint pixels to serve as alignment targets for drift pixels. They undergo categorical selection within the Adaptive Prototype Generation (APG) module to ensure their semantic richness. Compared with previous prototype-based methods, our prototypes embed rich local distribution information besides global distribution information, denoted as local prototypes. Hence, the drift pixels in each local distribution undergo asymmetric pushing toward the local prototypes in the Drift Pixels Asymmetric Contrast (DPA) module to harmonize pixels intra- and inter-distribution, ultimately approximating the global distribution.

Extensive experiments of multiple learning settings, including unsupervised domain adaptation (UDA), and semi-/fully-supervised segmentation, validate the performance of PiXL. In the UDA setting, PiXL outperforms the state-of-the-art methods on GTA5 and SYNTHIA → Cityscapes, respectively. In the semi-supervised setting on the Cityscapes dataset, PiXL produces consistent and competitive performance, especially on rare labels. When only 3.3% images are labeled, PiXL outperforms the SOTA by a large margin. The competitive results in the fully supervised setting on the Cityscapes dataset also show the power of pure pixel learning. The visualization of the pixel feature distribution illustrates that PiXL produces intra-class compact and inter-class separable pixel features for each category.

Our contributions are as follows:

• We introduce the pixel learning scheme by treating each image as a local distribution of pixels. It then can comprehensively analyze and meticulously handle the pixel-level variance to enhance the per-pixel recognition capability of the segmentation model and integrate various segmentation tasks under a unified framework.

• We propose the PiXL framework, which segregates pixels within a given local distribution into sub-domains: joint pixels and drift pixels. Then, the PiXL employs an asymmetric alignment approach to align drift pixels with the joint pixels, effectively addressing pixel-level variance in a divide-and-conquer manner.

● Extensive experiments confirm PiXL's performance, especially demonstrating promising results in label-scarce settings. The t-SNE visualization of the pixel embedding space further illustrates that PiXL attains enhanced intra-class compactness and inter-class distinctiveness in pixel features.

## 2 RELATED WORK

**Semantic segmentation using pixel features.** Semantic segmentation requires per-pixel classification, while previous image-level (Wang et al., 2020; Tsai et al., 2018) methods were limited to rough pixel recognition. Recent efforts strive to advance semantic segmentation to achieve a more refined level of granularity. In unsupervised or self-supervised learning, Xie et al. (2021c) adopted pixel-level contrastive learning as a pre-training task. In semi-supervised learning, Alonso et al. (2021) aligned the per-pixel feature to high-quality memory bank pixels. In weakly-supervised learning, Ahn & Kwak (2018); Du et al. (2022) extracted the pixel-level feature under supervision from CAMs (Zhou et al., 2016). Han et al. (2023) enforced pixel- and class-wise consistency constraints across various perspectives of 3D medical images. In fully supervised learning, Wang et al. (2021b) explored forming contrastive pairs with the cross-image pixels. In Ma et al. (2023), an image was considered as disorganized points, and clustering was employed to extract features, enhancing the model's interpretability. In unsupervised domain adaptation, Vayyat et al. (2022) conducted contrastive learning on pixels from multi-resolution images in CLUDA. Xie et al. (2023) unified different forms of pixel contrastive learning and added Gaussian distribution prior in SePiCo. However, few of these prior methods conducted an exhaustive analysis of pixel-level variance or devised learning strategies at a per-pixel level. Our PiXL, on the other hand, delves into pixel-level variance and tailors a pixel-learning scheme to address it.

**Domain adaptation.** Most studies in domain adaptation, including UDA, have predominantly emphasized addressing inter-domain variance (Zhang et al., 2021; Vu et al., 2019). However, the significance of intra-domain variance and even intra-distribution variance is equally noteworthy but has been rarely explored. Pan et al. (2020) proposed IntraDA to explicitly conduct intra-domain adaptation. Cai et al. (2022) extended IntraDA to an iterative adaptation manner. Yan et al. (2021) proposed PixIntraDA to conduct intra-domain alignment at the pixel level. Whereas the lack of comprehensive investigation of variance at the pixel level hampers addressing the issue. To bridge the gap, we analyze the causes of pixel-level variance in Fig. 1. Besides, we approach the pixel-level variance based on the pixel learning scheme in a divide-and-conquer manner to align drift pixels in local distribution to joint pixels following global distribution.

**Contrastive learning.** Contrastive learning is adopted either as a pretext task for pre-training (He et al., 2020; Wu et al., 2018; Zhao et al., 2021) or a plug-and-play module in a model (Xie et al., 2023; Vayyat et al., 2022) to refine feature representation. Utilizing the widely employed InfoNCE loss (Oord et al., 2018), features tend to approach positive samples while moving away from negative ones. In early works (Zhang et al., 2022; Chen et al., 2020), contrastive learning was used in image-level learning scenarios. Recently, some works attempted to apply contrastive learning at region (Xie et al., 2021a) or pixel (Xie et al., 2023; Wang et al., 2021b) level. Xie et al. (2021a) introduced patch-level contrastive learning by adding patch-level InfoNCE loss. Wang et al. (2022) applied contrastive learning at the pixel level with carefully selected contrastive pairs. Vayyat et al. (2022) introduced an explicit pixel-to-pixel contrastive manner in UDA. Nevertheless, these previous works lacked further consideration in contrastive mechanism design and appropriate positive sample selection, which could potentially cause adverse effects and misleading signals. Contrarily, PiXL develops an asymmetric contrast mechanism inspired by Yu et al. (2022) and collaborates with PSP and APG modules to guarantee the reliability of positive pixels and accurate alignment at the pixel level.

## 3 METHOD

### 3.1 PIXEL LEARNING

We first introduce the pixel learning scheme, where each image $X$ is construed as a local distribution of pixels sampled from the global pixel distribution of each category according to equation 1,

$$(X, Y) := \{(x_j, y_j) | j = 1, \cdots, N; \ y_j = 1, \cdots, C\} \tag{1}$$

$j$ represents the pixel index, and $N$ signifies an image's total number of pixels. For pixel features of each class $k$, let $\mathcal{G}_k$ denote the global distribution and $g_k$ indicate the local distribution. Consequently, feature maps $\boldsymbol{F}$ with their corresponding labels $\overline{Y}$ are considered as a distribution of labeled pixel features, as depicted in equation 2, where $\overline{N}$ denotes the number of pixel features in $\boldsymbol{F}$. While pixel features in $g_k$ are anticipated to follow $\mathcal{G}_k$, this is not always true.

$$(\boldsymbol{F}, \overline{Y}) \coloneqq g = \{(\boldsymbol{f}_j, \overline{y}_j) \,|\, j = 1, \cdots, \overline{N};\; y_j = 1, \cdots, C\} = \bigcup_{k=1}^{C} g_k, \tag{2}$$

where $g_k = \{(\boldsymbol{f}_j, \overline{y}_j) \,|\, \overline{y}_j = k\}$, $\boldsymbol{f}_j, \overline{y}_j$ are pixel feature and label.

For the sake of narrative, **image** is also referred to **distribution** in this work. The pixel learning scheme contributes to semantic segmentation in the following two aspects:

• Pixel learning elevates the granularity of variance analysis to the pixel level, concentrating on intra- and inter-distribution, pixel-level variance, especially emphasizing the intra-image pixel variance.

• Pixel learning serves as a unified scheme in semantic segmentation, entailing a continuous intra- and inter-distribution pixel-level domain adaptation, i.e., the ongoing alignment of pixel features from two sampled distributions could ultimately address tasks like UDA, semi- and fully-supervised semantic segmentation.

### 3.2 OVERVIEW OF PiXL FRAMEWORK

Based upon principles of pixel learning, we construct a pixel learning framework called PiXL, consisting of four modules, i.e., i). Multiple Resolution Feature Extraction, ii). Pixel Level Sub-Domain Partition (PSP), iii). Adaptive Prototype Generation (APG), iv). Drift Pixels Alignment (DPA).

At each training step in PiXL, the learning process proceeds as follows:

• Two images are sampled to construct a local pixel feature distribution $g$ by obtaining the features of pixels from the extracted feature map $\boldsymbol{F}$ and the corresponding label $\overline{Y}$. We employ a multi-resolution input strategy to incorporate more comprehensive semantic information.

• The PSP module partitions the pixel features within the local distribution into two categories: $\hat{\boldsymbol{F}} = \{\hat{\boldsymbol{f}}_1, \cdots, \hat{\boldsymbol{f}}_{N_j}\}$ representing joint pixel features assumed to follow the global feature distribution of each class, and $\tilde{\boldsymbol{F}} = \{\tilde{\boldsymbol{f}}_1, \cdots, \tilde{\boldsymbol{f}}_{N_d}\}$ denoting drift pixel features considered to deviate from those distributions.

• The APG module generates multi-resolution prototypes for each category using the joint pixel features and adaptively selects the semantic richness prototype as the alignment target for each category.

• The drift pixels are aligned to the categorical global pixel feature distribution by pulling them towards prototypes of the same class while simultaneously pushing them away from other prototypes in the DPA module.

### 3.3 MULTIPLE RESOLUTION FEATURE EXTRACTION

We employ a multiple-resolution input strategy to guarantee the semantic richness in features and the model's robustness to varied resolutions of images, inspired by Hoyer et al. (2022b). Specifically, PiXL cuts out a high-resolution part $X^H$ from image $X$ while resizing $X$ to the same size of $X^H$ to form a low-resolution image $X^L$, providing abundant details and sufficient context information respectively. The extracted feature maps $\boldsymbol{F}^L$ and $\boldsymbol{F}^H$ from $X^L$ and $X^H$, collaborated with corresponding labels $\overline{Y}^L$ and $\overline{Y}^H$, constitute the local distribution of pixel features given by equation 3.

$$g = g^L \cup g^H = \{(\boldsymbol{f}_j^L, \overline{y}_j) \,|\, j = 1, \cdots, \overline{N}_L\} \cup \{(\boldsymbol{f}_j^H, \overline{y}_j) \,|\, j = 1, \cdots, \overline{N}_H\} \tag{3}$$

### 3.4 PIXEL LEVEL SUB-DOMAIN PARTITION

PiXL employs entropy as the criteria to segregate the pixel features in $g$ into joint pixels and drift pixels. The entropy $h$ of pixel feature $\boldsymbol{f}$ is determined by its predicted probability distribution. To

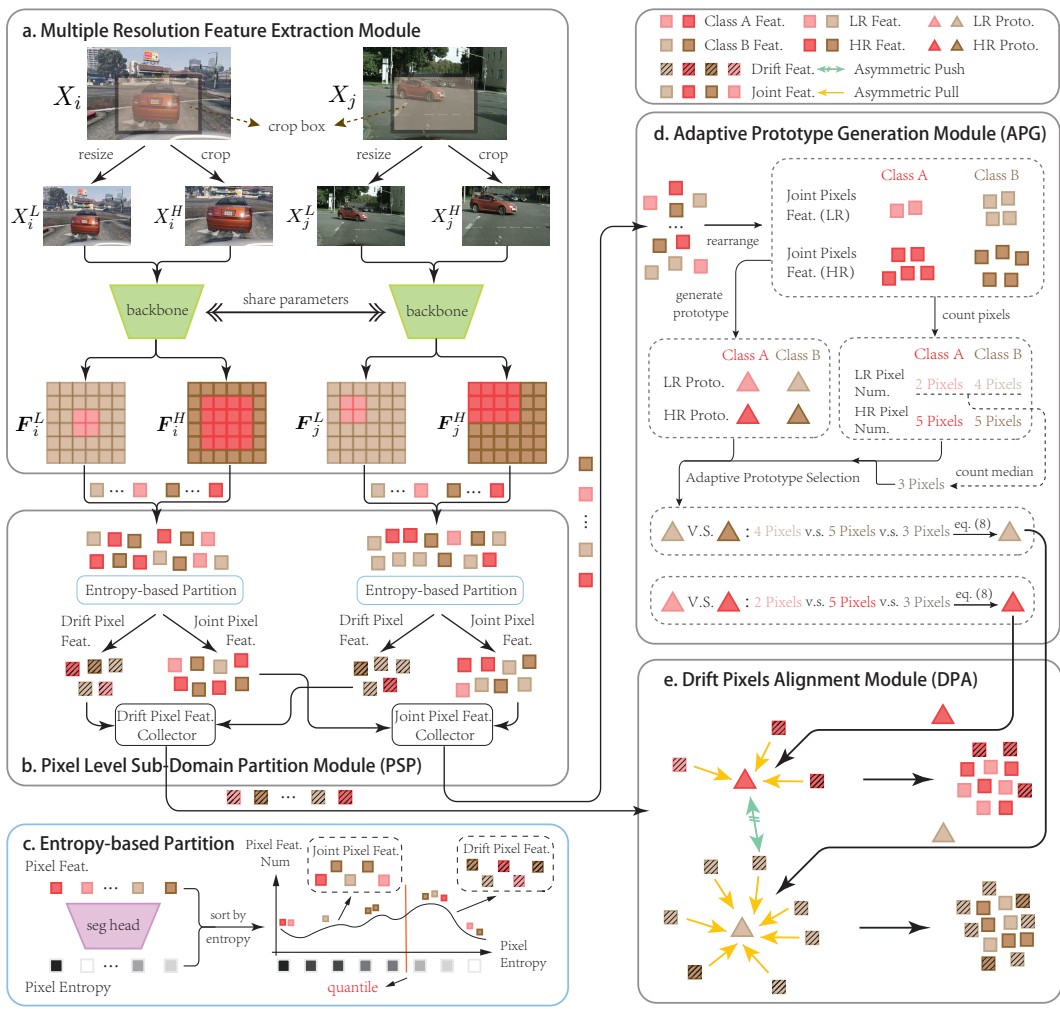

Figure 2: An overview of PiXL framework. At each training step, two images are sampled and fed into the subsequent modules. Multiple Resolution Feature Extraction Module extracts features from low- and high-resolution images. Pixel-Level Sub-Domain Partition Module arranges pixels from each image into joint pixels and drift pixels based on entropy. The Adaptive Prototype Generation Module selects prototypes from different resolutions to balance context and details. Drift Pixels Alignment Module aligns drift pixels to local prototypes.

pursue a suitable partition of pixel features, low-resolution and high-resolution pixel features are concatenated and then sorted according to their per-pixel entropy. Subsequently, PiXL determines the threshold $\epsilon$ using the $1 - \eta$ quantile of the sorted pixel entropy to conduct partition according to equation 4. It should be noted that pixel partition is performed separately for each distribution to ensure fairness, considering the entropy gap across images.

$$\hat{F} = \{\hat{f}_j | h_j < \epsilon\}, \ \tilde{F} = \{\tilde{f}_j | h_j \geq \epsilon\}, \ j = 1, \cdots, \overline{N}_L + \overline{N}_H. \tag{4}$$

## 3.5 DRIFT PIXELS ALIGNMENT

**Local prototypes.** To conduct intra- and inter-distribution alignment on pixels from two sampled distributions, local prototypes for each category are calculated on joint pixels. Given joint pixels from distribution $i$ and distribution $j$, local prototype for category $k$ is given by:

$$\boldsymbol{p}^{(k)} = \frac{1}{|\hat{\boldsymbol{F}}_i^{(k)} \cup \hat{\boldsymbol{F}}_j^{(k)}|} \sum \hat{\boldsymbol{f}}_t, \quad \text{where } \hat{\boldsymbol{f}}_t \in \hat{\boldsymbol{F}}_i^{(k)} \cup \hat{\boldsymbol{F}}_j^{(k)}. \tag{5}$$

The $\hat{\boldsymbol{F}}_i^{(k)}$ and $\hat{\boldsymbol{F}}_j^{(k)}$ refer to feature of joint pixels belongs to category $k$ from distribution $i$ and distribution $j$ respectively.

**Drift pixels asymmetric contrast.** Contrastive learning is employed to align pixel features intra- and inter-distribution. We design an asymmetric alignment strategy to push drift pixels to local prototypes $\boldsymbol{p}$ and suppress the misleading signals from drift pixels. Specifically, given a drift pixel $\tilde{\boldsymbol{f}}_t$ affiliated to category $k$ as the anchor. The positive sample is the local prototype $\boldsymbol{p}^{(k)}$ while other local prototypes $\{\boldsymbol{p}^{(j)}|j=1,\cdots C, j\neq k\}$ are negative samples. Thus, for the single drift pixel $\tilde{\boldsymbol{f}}_t$, the asymmetric contrast loss is given as follows:

$$\mathcal{L}_t^{DPA} = \mathcal{L}_{InfoNCE}\left(\tilde{\boldsymbol{f}}_t, \mathscr{P}_t, \mathcal{N}_t\right) = \frac{1}{|\mathscr{P}_t|}\sum_{\tilde{\boldsymbol{f}}_t\in\mathscr{P}_t}\left[-\log\left(\frac{e^{(\tilde{\boldsymbol{f}}_t\cdot\boldsymbol{p}^{(k)}/\tau)}}{e^{(\tilde{\boldsymbol{f}}_t\cdot\boldsymbol{p}^{(k)}/\tau)}+\sum_{\boldsymbol{p}^{(j)}\in\mathcal{N}_t}e^{(\tilde{\boldsymbol{f}}_t\cdot\boldsymbol{p}^{(j)}/\tau)}}\right)\right]$$

**where** $\mathscr{P}_t=\{\boldsymbol{p}^{(k)}\}, \mathcal{N}_t=\{\boldsymbol{p}^{(j)}|j=1,\cdots,C, j\neq k\}, \tau$ is temperature parameter.

(6)

PiXL stops the gradient accumulation of prototypes $\boldsymbol{p}$ in the DPA module to realize the asymmetric contrast. The asymmetric alignment loss to address all drift pixels from two distributions is given by equation 7.

$$\mathcal{L}_{DPA}(\tilde{\boldsymbol{F}}_i, \tilde{\boldsymbol{F}}_j) = \frac{1}{|\tilde{\boldsymbol{F}}_i\cup\tilde{\boldsymbol{F}}_j|}\sum_{\tilde{\boldsymbol{f}}_t}\mathcal{L}_t^{DPA}.$$

(7)

### 3.6 ADAPTIVE PROTOTYPE GENERATION

Considering the semantic variation in the features extracted from images of different resolutions, the local prototypes calculated from them should share complementary information. For common classes, like *sky*, *road*, and *building*, context information gets compromised in high-resolution features $\boldsymbol{F}^H$. On the contrary, the low-resolution features $\boldsymbol{F}^L$ lack sufficient details. Thus, we propose an adaptive local prototype selection module to choose the most meaningful pixels for each category to form local prototypes. As shown in Fig. 2, the number of pixels embedded in each prototype is also quantified simultaneously during the generation of multi-resolution prototypes, which is referred to as $\{(\boldsymbol{p}^{(k)}, N^{(k)})|k=1,\cdots,C\}$. We sort these low-resolution prototypes $\boldsymbol{p}_L^{(k)}$ according to their corresponding number of pixels $N_L^{(k)}$ and compute the median of $N_L^{(k)}$, denoted as $m$. For a class $k$, the adaptive selection of the prototype is given by equation 8.

$$\boldsymbol{p}^{(k)} = \begin{cases} \boldsymbol{p}_L^{(k)}, N_L^{(k)}\geq m, \\ \boldsymbol{p}_L^{(k)}, N_L^{(k)}<m \text{ and } N_H^{(k)}<N_L^{(k)}, \\ \boldsymbol{p}_H^{(k)}, \text{ otherwise.} \end{cases}$$

(8)

### 3.7 LOSS

In addition to the $\mathcal{L}_{DPA}$ loss, cross-entropy (CE) loss $\mathcal{L}_{CE}^*$ and a Thing-Class ImageNet Feature Distance loss $\mathcal{L}_{FD}^*$ following Hoyer et al. (2022a;b); Vayyat et al. (2022) are also employed. The complete loss is summarized in equation 9, where $\lambda_{FD}$ is the weight of $\mathcal{L}_{FD}^*$. Please refer to the Appx. § A.3 for details of $\mathcal{L}_{CE}^*$ and $\mathcal{L}_{FD}^*$.

$$\mathcal{L}_{PiXL} = \mathcal{L}_{CE}^* + \lambda_{FD}\mathcal{L}_{FD}^* + \mathcal{L}_{DPA}.$$

(9)

## 4 EXPERIMENTS

### 4.1 DATASET

**GTA5.** A large-scale synthetic dataset contains 24,966 annotated images. **SYNTHIA.** A collection of generated urban images including 9,400 images. **Cityscapes.** A real-world street scene dataset contains 2,975 training images and 500 validation images.

## 4.2 TASK SETTING

**Fully supervised learning.** We use all the training images from Cityscapes to train our PiXL and evaluate that on the corresponding validation part. **Semi-supervised learning.** We randomly select 1/4, 1/8, and 1/30 labeled images from the Cityscapes training set while utilizing the left images as unlabeled images to train PiXL. Performance is reported on the validation part. **Unsupervised domain adaptation.** We evaluate our PiXL on GTA5→Cityscapes and SYNTHIA →Cityscapes. mIoU is utilized to evaluate the model's performance in these tasks, which is calculated by the average of the intersection over the union in each category.

## 4.3 IMPLEMENTATION DETAILS

**Network architecture.** The HRDA model (Hoyer et al., 2022b) is adopted as the baseline, which consists of a MiT-B5 encoder (Xie et al., 2021b) and a feature fusion decoder from DAFormer (Hoyer et al., 2022a). We further validate the effectiveness and plug-and-play property of PiXL on DeepLabV3+ (Chen et al., 2018) backbone, referring to Appx. § A.2 for more details.

**Training.** We follow the Vayyat et al. (2022) training strategies and parameters, i.e. a batch size of 2, the optimizer is AdamW with a learning rate of $6 \times 10^{-5}$ for the encoder and $6 \times 10^{-4}$ for the decoder with linear warmup policy, DACS data augmentation, rare class sampling strategy on labeled training data, the self-training method for unlabeled training data. In UDA and SSL settings, we employ the Exponential Moving Average (EMA) updated teacher model to generate the pseudo labels for unlabeled images, as shown in Fig. 5. Moreover, we set the training epoch to 60000, set the $\eta$ to 0.2 initially, and decrease it linearly until $\eta = 0.001$ during training. The model is trained on a single Tesla V100 with 32 GB memory. More details are provided in Appx. § A.4.

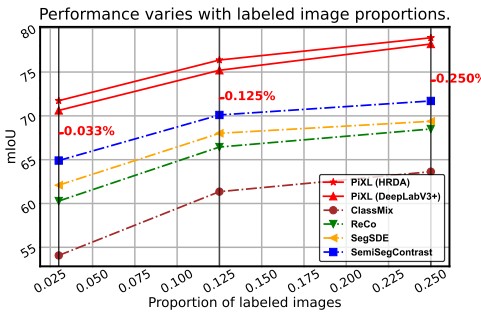

Figure 3: Performance on different proportions of labeled images.

Table 1: Comparison with previous methods in the semi-supervised setting on Cityscapes.

| Methods | Image Size | 1/30(100) | 1/8(372) | 1/4(744) |
|---|---|---|---|---|
| **ClassMix** Olsson et al. (2021) | 512×1024 | 54.07 | 61.35 | 63.63 |
| **SemiSegContrast** Alonso et al. (2021) | 512×512 | 64.90 | 70.10 | 71.70 |
| **CCT** Ouali et al. (2020) | 512×1024 | - | 74.12 | 75.99 |
| **GCT** Ke et al. (2020) | 769×769 | - | 72.66 | 76.11 |
| **MT** Tarvainen & Valpola (2017) | 800×800 | - | 73.71 | 76.53 |
| **AEL** Hu et al. (2021) | 769×769 | - | **77.90** | 79.01 |
| **U2PL** Wang et al. (2022) | 769×769 | - | 76.48 | 78.51 |
| **CPS** Chen et al. (2021) | 800×800 | - | 77.62 | **79.21** |
| **ReCo** Liu et al. (2021a) | 512×512 | 60.28 | 66.44 | 68.50 |
| **SegSDE** Hoyer et al. (2021) | 512×512 | 62.09 | 68.01 | 69.38 |
| **PiXL (DeepLabV3+)** | 512×512 | 70.62 | 75.20 | 78.20 |
| **PiXL (HRDA)** | 512×512 | **71.73** | 76.37 | 78.91 |

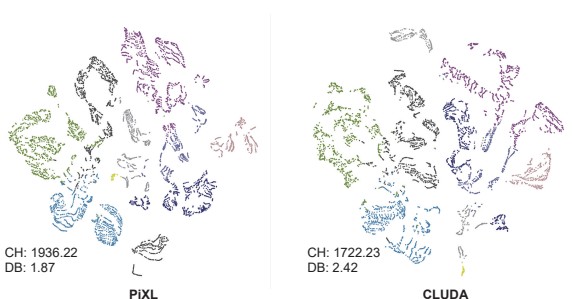

Figure 4: t-SNE of pixel features in CLUDA and PiXL.

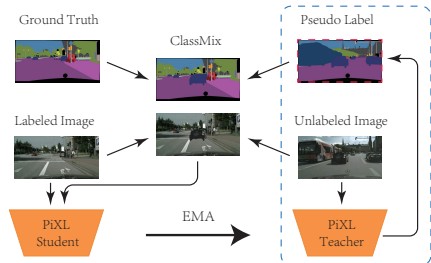

Figure 5: Pseudo labels generation in PiXL

Table 2: Comparison with previous methods in UDA setting on GTA5, SYNTHIA → Cityscapes.

| Methods | Road | S.Walk | Build. | Wall | Fence | Pole | Tr.Light | Sign | Veget. | Terrain | Sky | Person | Rider | Car | Truck | Bus | Train | M.Bike | Bike | mIoU |
|---|---|---|---|---|---|---|---|---|---|---|---|---|---|---|---|---|---|---|---|---|
| **GTA5 → Cityscapes** | | | | | | | | | | | | | | | | | | | | |
| **AdaptSeg** Tsai et al. (2018) | 86.5 | 25.9 | 79.8 | 22.1 | 20.0 | 23.6 | 33.1 | 21.8 | 81.8 | 25.9 | 75.9 | 57.3 | 26.2 | 76.3 | 29.8 | 32.1 | 7.2 | 29.5 | 32.5 | 41.4 |
| **CBST** Zou et al. (2018) | 91.8 | 53.5 | 80.5 | 32.7 | 21.0 | 34.0 | 28.9 | 20.4 | 83.9 | 34.2 | 80.9 | 53.1 | 24.0 | 82.7 | 30.3 | 35.9 | 16.0 | 25.9 | 42.8 | 45.9 |
| **DACS** Tranheden et al. (2021) | 89.9 | 39.7 | 87.9 | 30.7 | 39.5 | 38.5 | 46.4 | 52.8 | 88.0 | 44.0 | 88.8 | 67.2 | 35.8 | 84.5 | 45.7 | 50.2 | 0.0 | 27.3 | 34.0 | 52.1 |
| **CorDA** Wang et al. (2021a) | 94.7 | 63.1 | 87.6 | 30.7 | 40.6 | 40.2 | 47.8 | 51.6 | 87.6 | 47.0 | 89.7 | 66.7 | 35.9 | 90.2 | 48.9 | 57.5 | 0.0 | 39.8 | 56.0 | 56.6 |
| **BAPA** Liu et al. (2021b) | 94.4 | 61.0 | 88.0 | 26.8 | 39.9 | 38.3 | 46.1 | 55.3 | 87.8 | 46.1 | 89.4 | 68.8 | 40.0 | 90.2 | 60.4 | 59.0 | 0.0 | 45.1 | 54.2 | 57.4 |
| **ProDA** Zhang et al. (2021) | 87.8 | 56.0 | 79.7 | 46.3 | 44.8 | 45.6 | 53.5 | 53.5 | 88.6 | 45.2 | 82.1 | 70.7 | 39.2 | 88.8 | 45.5 | 59.4 | 1.0 | 48.9 | 56.4 | 57.5 |
| **DAFormer** Hoyer et al. (2022a) | 95.7 | 70.2 | 89.4 | 53.5 | 48.1 | 49.6 | 55.8 | 59.4 | 89.9 | 47.9 | 92.5 | 72.2 | 44.7 | 92.3 | 74.5 | 78.2 | 65.1 | 55.9 | 61.8 | 68.3 |
| **SePiCo** Xie et al. (2023) | 96.9 | 76.7 | 89.7 | 55.5 | 49.5 | 53.2 | 60.0 | 64.5 | 90.2 | 50.3 | 90.8 | 74.5 | 44.2 | 93.3 | 77.0 | 79.5 | 63.6 | 61.0 | 65.3 | 70.3 |
| **HRDA** Hoyer et al. (2022b) | 96.4 | 74.4 | 91.0 | **61.6** | 51.5 | 57.1 | 63.9 | 69.3 | 91.3 | 48.4 | 94.2 | 79.0 | 52.9 | 93.9 | **84.1** | 85.7 | 75.9 | 63.9 | 67.5 | 73.8 |
| **CLUDA** Vayyat et al. (2022) | **97.1** | **78.0** | 91.0 | 60.3 | **55.3** | 56.3 | 64.3 | **71.5** | 91.2 | 51.1 | 94.7 | 78.4 | 52.9 | 94.5 | 82.8 | 86.5 | 73.0 | 64.2 | **69.7** | 74.4 |
| **PiXL** | 97.0 | 77.6 | 91.1 | 59.9 | 54.1 | 57.2 | 64.8 | 69.1 | 91.5 | 51.8 | 94.8 | 80.5 | 57.3 | 94.6 | 83.8 | 88.7 | 78.0 | 65.6 | 67.8 | 75.0 |
| **SYNTHIA → Cityscapes** | | | | | | | | | | | | | | | | | | | | |
| **AdaptSeg** Tsai et al. (2018) | 79.2 | 37.2 | 78.8 | - | - | - | 9.9 | 10.5 | 78.2 | - | 80.5 | 53.5 | 19.6 | 67.0 | - | 29.5 | - | 21.6 | 31.3 | 37.2 |
| **CBST** Zou et al. (2018) | 68.0 | 29.9 | 76.3 | 10.8 | 1.4 | 33.9 | 22.8 | 29.5 | 77.6 | - | 78.3 | 60.6 | 28.3 | 81.6 | - | 23.5 | - | 18.8 | 39.8 | 42.6 |
| **DACS** Tranheden et al. (2021) | 80.6 | 25.1 | 81.9 | 21.5 | 2.9 | 37.2 | 22.7 | 24.0 | 83.7 | - | 90.8 | 67.5 | 38.3 | 82.9 | - | 38.9 | - | 28.5 | 47.6 | 48.3 |
| **CorDA** Wang et al. (2021a) | **93.3** | **61.6** | 85.3 | 19.6 | 5.1 | 37.8 | 36.6 | 42.8 | 84.9 | - | 90.4 | 69.7 | 41.8 | 85.6 | - | 38.4 | - | 32.6 | 53.9 | 55.0 |
| **BAPA** Liu et al. (2021b) | 91.7 | 53.8 | 83.9 | 22.4 | 0.8 | 34.9 | 30.5 | 42.8 | 86.8 | - | 88.2 | 66.0 | 34.1 | 86.6 | - | 51.3 | - | 29.4 | 50.5 | 53.3 |
| **ProDA** Zhang et al. (2021) | 87.8 | 45.7 | 84.6 | 37.1 | 0.6 | 44.0 | 54.6 | 37.0 | 88.1 | — | 84.4 | 74.2 | 24.3 | 88.2 | — | 51.1 | — | 40.5 | 45.6 | 55.5 |
| **DAFormer** Hoyer et al. (2022a) | 84.0 | 40.7 | 88.4 | 41.5 | 6.5 | 50.0 | 55.0 | 54.6 | 86.0 | - | 89.8 | 73.2 | 48.2 | 87.2 | - | 53.2 | - | 53.9 | 61.7 | 60.9 |
| **SePiCo** Xie et al. (2023) | 87.0 | 52.6 | 88.5 | 40.6 | **10.6** | 49.8 | 57.0 | 55.4 | 86.8 | - | 86.2 | 75.4 | 52.7 | **92.4** | - | **78.9** | - | 53.0 | 62.6 | 64.3 |
| **HRDA** Hoyer et al. (2022b) | 85.2 | 47.7 | 88.8 | **49.5** | 4.8 | 57.2 | 65.7 | 60.9 | 85.3 | - | 92.9 | 79.4 | 52.8 | 89.0 | - | 64.7 | - | 63.9 | 64.9 | 65.8 |
| **CLUDA** Vayyat et al. (2022) | 87.7 | 46.9 | **90.2** | 49.0 | 7.9 | **59.5** | **66.9** | 58.5 | **88.3** | - | 94.6 | 80.1 | **57.1** | 89.8 | - | 68.2 | - | 65.5 | **65.8** | 66.8 |
| **PiXL** | 89.9 | 53.8 | 90.1 | **52.8** | 7.1 | 58.8 | 49.9 | 63.5 | 88.4 | - | 94.6 | 80.5 | 55.9 | 90.8 | - | 68.1 | - | 67.0 | 62.9 | 67.1 |

Table 3: Comparison with previous methods in the fully supervised setting on Cityscapes.

| Methods | Image Size | Cityscapes |
|---|---|---|
| **HRNetV2** Wang et al. (2020) | 512×1024 | 81.10 |
| **HRViT-b1** Gu et al. (2022) | 1024×1024 | 81.63 |
| **ContrastiveSeg** Wang et al. (2021b) | 512×1024 | 82.40 |
| **HRViT-b2** Gu et al. (2022) | 1024×1024 | 82.81 |
| **SegFormer** Xie et al. (2021) | 1024×1024 | 84.00 |
| **Mask2Former** Cheng et al. (2022) | 512×1024 | 84.50 |
| **SeMask** Jain et al. (2021) | 768×768 | **84.98** |
| **PiXL** | 512×512 | 82.74 |

Table 4: Component ablation of PiXL.

| Methods | APG | DPA | mIoU | ↑ |
|---|---|---|---|---|
| **baseline** | | | 73.8 | |
| **PiXL** | | ✓ | 74.7 | **+0.9** |
| **PiXL** | ✓ | ✓ | **75.0** | **+1.2** |

Table 5: Quantile ablation of PiXL.

| $\eta$ | 0.1 | 0.2 | 0.4 | 0.6 | 0.8 |
|---|---|---|---|---|---|
| **mIoU** | 74.8 | **75.0** | **75.0** | 74.9 | 74.1 |

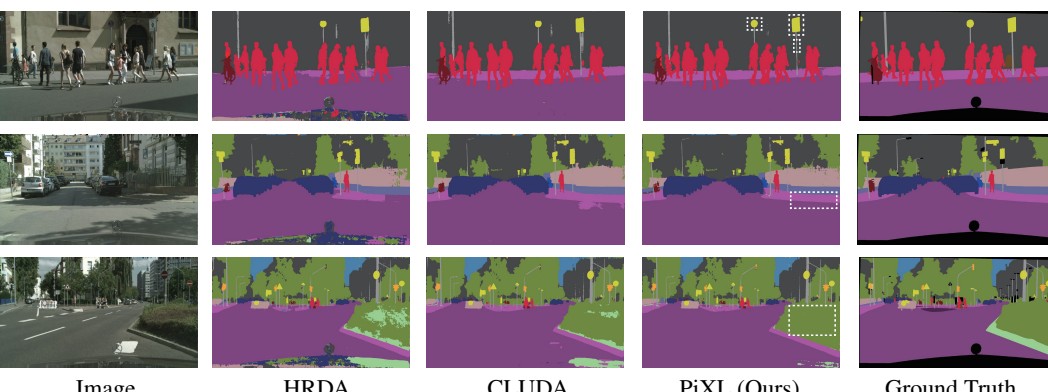

| Image | HRDA | CLUDA | PiXL (Ours) | Ground Truth |

Figure 6: Qualitative comparison with baseline and SOTA methods in UDA setting on GTA5 → Cityscapes. PiXL performs better on boundary pixels and internal pixels.

## 4.4 RESULTS

**Unsupervised domain adaptation.** In the UDA setting, our PiXL outperforms the baseline model HRDA and achieves commensurate performance with the SOTA. In GTA5 → Cityscapes and SYN-THIA → Cityscapes, our model PiXL surpasses the baseline by a margin of 1.2% and 1.3% respectively in Tab. 2. Compared with the SOTA model CLUDA, our method outperforms it by a margin

of 0.6% and 0.3%. Specifically, in GTA5 → Cityscapes, PiXL obviously improves the performance on rare classes, like *pole*, *motorbike*, *etc*. Moreover, the performance improvement is also consistent in SYNTHIA → Cityscapes. In the common class pixels, PiXL also achieves comparable results or surpasses the previous works, like *vegetation*, *road*, *sidewalk*, *etc*. Fig. 6 provides more details, depicting that PiXL performs better in internal and boundary pixels. Experiments conducted in the UDA setting confirm the effectiveness of PiXL, implying that the pixel learning scheme produces more consistent pixel features and mitigates the domain gap at the pixel level.

**Semi-supervised semantic segmentation.** We evaluate PiXL on DeepLabV3+ (Chen et al., 2020) and HRDA (Hoyer et al., 2022b) to verify its plug-and-play property. PiXL performs competitively with the SOTA models as summarized in Tab. 1, especially in 3.3% images labeled setting, where PiXL (DeepLabV3+), PiXL (HRDA) surpass the SOTA by 5.72% and 6.83% respectively. Fig. 3 further confirms PiXL maintains its excellent and robust performance as the labeled image ratio varies. The qualitative analyses are available in Appx. § A.5. Experiments on semi-supervised segmentation demonstrate PiXL's ability to thrive in label-efficient scenarios, which further highlights the strengths of pixel learning.

**Fully supervised semantic segmentation.** As depicted in Tab. 3, PiXL surpasses HRNetV2 and HRViT-b1, which concentrate on multi-resolution feature mining. PiXL exhibits competitive performance compared with the SOTA methods trained on higher resolution images, such as $768 \times 768$, $512 \times 1024$, and even $1024 \times 1024$. Notably, PiXL achieves this level of performance while being trained on $512 \times 512$ images. Although images of lower resolution and smaller fields of view may potentially deteriorate pixel feature representation, which exacerbates the bias of the local distribution, the pixel learning scheme effectively mitigates these influences to some extent and continues to deliver competitive performance. The qualitative analyses are provided in Appx. § A.5.

**Ablation study.** We conduct abundant ablation of PiXL in GTA5 → Cityscapes task. The ablation on components, including the DPA and APG modules, are summarized in Tab. 4. Compared with the baseline model HRDA, our drift pixels alignment module improves the performance by 0.9%, which verifies that the asymmetric contrast mechanism effectively addresses pixel feature variance and ultimately improves the model's per-pixel recognition capability. Adopting the adaptive selection in the APG module further improves the performance of PiXL to 75.0%, which outperforms the baseline by a margin of 1.2%. This underscores that the adaptive selection contributes to achieving a balance between details and contextual information encapsulated in the local prototypes, making them better alignment targets for drift pixels. Despite the ablation study of $\eta$ presented in Tab. 5 demonstrating robust improvement on the baseline performance 73.8%, $\eta$ still affects the model slightly. Considering the partition threshold $\epsilon$ is determined by $1 - \eta$ quantile, a larger $\eta$ means more pixels are drift pixels. The improvement is marginal when $\eta$ is 0.8, confirming that the drift pixels are indeed a small portion and tailored adjustment in PiXL refines each category's pixel features.

**Visualization.** We employ t-SNE to visualize the distribution of pixel features in the embedding space and compare the pixel features of PiXL and the SOTA method CLUDA. Fig. 4 illustrates that PiXL refines pixel feature representation with better inter-class separability and intra-class compactness. PiXL's performance on quantitative metrics Calinski-Harabasz Index (CH) Davies-Bouldin Index (DB), *i.e.*, 1936.22 and 1.87 while that of CLUDA is 1722.23 and 2.42, further confirming the above conclusion, as a larger CH and a smaller DB indicate better inter-class separability and intra-class compactness. Both visualization and quantitative analysis substantiate the improvement in pixel features achieved by PiXL.

## 5 CONCLUSION

In this work, we propose a novel pixel learning scheme to dive semantic segmentation models into the pixel level by treating an image as a distribution of pixels. This advocates addressing pixel-level variance to enhance the segmentation model's per-pixel recognition capability. We proposed PiXL, a pure pixel-level learning framework that executes pixel-level intra- and inter-distribution (image) alignment within the context of pixel learning. Extensive quantitative and qualitative experiments in various settings of semantic segmentation confirm the effectiveness of PiXL, demonstrating the feasibility and superiority of the pixel learning scheme, which deserves further exploration.

REPRODUCIBILITY STATEMENT

To guarantee the reproducibility and completeness of this paper, we devote the following efforts:

• We illustrate the model pipeline in Fig. 2 and Fig. 5, with detailed explanations provided in section § 3.2 to facilitate understanding.

• We provide abundant details regarding the model implementation and hyper-parameters choice in section § 4.3, Appx. § A.2, and Appx. § A.4.

• We comprehensively explain the loss functions in our work in Appx. § A.3.

• The code will be available upon acceptance.

ACKNOWLEDGMENTS

This work is partially supported by the National Natural Science Foundation of China (grant no. 62106101), and the Natural Science Foundation of Jiangsu Province (grant no. BK20210180). This work is also partially supported by the AI & AI for Science Project of Nanjing University.

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

## A  APPENDIX

### A.1  OVERVIEW

In this appendix, we provide further insights into the PiXL framework, including implementation details § A.2, loss functions § A.3, training and inference procedures § A.4, and supplementary qualitative analysis § A.5.

### A.2  IMPLEMENTATION DETAILS

We have implemented two variants of the PiXL framework utilizing different backbones and decoders, as depicted in Tab 6. All the backbones have been pre-trained on ImageNet-1K.

Table 6: Variants of PiXL. † indicates the inclusion of modifications.

| Framework | Backbone | Decoder |
|---|---|---|
| **PiXL (DeepLabV3+)** | ResNet101 | DeepLabV3+ Decoder† |
| **PiXL (HRDA)** | MiT-B5 | DAFormer Decoder |

**PiXL (HRDA)**   We adopt the MiT-B5 as the backbone and employ a context-aware feature fusion decoder proposed by Hoyer et al. (2022a). The MiT-B5 extracts multi-scale features from input images, which are passed through an Atrous Spatial Pyramid Pooling (ASPP) (Chen et al., 2017). This module generates the final feature representation for per-pixel prediction and constitutes the pixel features used for pixel learning. The feature dimension $C$ of each pixel is 256. The attention-based multi-scale prediction fusion mechanism is also applied in our PiXL following Hoyer et al. (2022b).

**PiXL (DeepLabV3+)**   To validate the plug-and-play property of PiXL, we have also implemented a CNN-based variant, which comprises a ResNet101 (He et al., 2016) encoder and a DeepLabV3+ head (Chen et al., 2018). In addition to enriching pixel features by combining the low-level features with the fused features, we modify the input of the ASPP head by aligning and stacking features from all four encoder blocks, similar to the approach employed in the DAFormer head.

### A.3  LOSS FUNCTIONS

According to § 3.7, the loss function in PiXL encompasses cross-entropy loss $\mathcal{L}_{CE}^*$, Thing-Class ImageNet Feature Distance loss $\mathcal{L}_{FD}^*$ and our Drift Pixel Alignment loss $\mathcal{L}_{DPA}$, as detailed in § 3.5.

**Cross-entropy loss**   Given feature map $\boldsymbol{F}$, its cross-entropy loss is given following equation 10, where $p_{t_k}$ denotes the probability of pixel feature $\boldsymbol{f}_t$ belongs to class $k$. $\overline{y}_t$ signifies the label associated with feature $\boldsymbol{f}_t$.

$$\mathcal{L}_{CE}(\boldsymbol{F}) = -\frac{1}{|\boldsymbol{F}|} \sum_{t=1,\cdots,|\boldsymbol{F}|} w_t \sum_{k=1,\cdots,C} \mathbb{I}_{[\overline{y}_t=k]} \log p_{t_k} \tag{10}$$

$w_t$ indicates the per-pixel cross-entropy loss weight, set to 1.0 when the label is available. Otherwise, $w_t$ is determined by the confidence of pseudo labels to relieve noise from them. In the context where $\boldsymbol{F}_j \in \mathbb{R}^{h \times w \times C}$ is obtained from an unlabeled image $X_j$, we use $\overline{Y}_j \in \mathbb{R}^{h \times w}$ to denote the pseudo label predicted by the EMA teacher. Additionally, $\boldsymbol{M}_j \in \mathbb{R}^{h \times w}$ is employed to represent the confidence of each pixel, with each value corresponding to the maximum value within each pixel's probability distribution. For calculating $w_t$ for each pixel in $\boldsymbol{F}_j$, we utilize the formula provided in equation 11, with $\delta$ set to 0.968.

$$w_t = \frac{\sum_{x=1}^{h} \sum_{y=1}^{w} [\boldsymbol{M}_j(x,y) > \delta]}{h \times w}. \tag{11}$$

Since we sample two distributions at each training step and employ a multi-resolution input strategy, the comprehensive cross-entropy loss is defined as follows:

$$\mathcal{L}_{CE}^* = \mathcal{L}_{CE}(\boldsymbol{F}_i^L) + \mathcal{L}_{CE}(\boldsymbol{F}_j^L) + \lambda_H \mathcal{L}_{CE}(\boldsymbol{F}_i^H) + \lambda_H \mathcal{L}_{CE}(\boldsymbol{F}_j^H). \tag{12}$$

The value of $\lambda_H$ is set to 0.1 in equation 12.

**Thing-Class ImageNet Feature Distance loss**   The Thing-Class ImageNet Feature Distance loss, referred to as $\mathcal{L}_{FD}$, is employed to leverage the recognition capability of an ImageNet pre-trained backbone for training regularization purposes, following Hoyer et al. (2022a;b); Vayyat et al. (2022). Let $\mathscr{F}'$ represent the ImageNet pre-trained backbone and $\dot{\boldsymbol{F}}'$ denote the last layer features extracted from $\mathscr{F}'$. Similarly, $\mathscr{F}$ and $\dot{\boldsymbol{F}}$ represent the backbone and its corresponding last layer features from PiXL. The distance between each pixel feature $\dot{\boldsymbol{f}}_j$ in $\dot{\boldsymbol{F}}$ and its corresponding feature $\dot{\boldsymbol{f}}_j'$ in $\dot{\boldsymbol{F}}'$ is calculated as follows:

$$d_j = \left\| \dot{\boldsymbol{f}}_j - \dot{\boldsymbol{f}}_j' \right\|_2 \tag{13}$$

As illustrated in Hoyer et al. (2022a;b); Vayyat et al. (2022), the $\mathcal{L}_{FD}$ is applied to thing classes (Caesar et al., 2018), denoted as $C_{things}$. A mask $I$ is generated to distinguish between things and stuff, where each element $I_j$ indicates whether feature $\boldsymbol{f}_j \in C_{things}$. Consequently, the $\mathcal{L}_{FD}$ on features from ImageNet-pretrained backbone $\mathscr{F}'$ and PiXL backbone $\mathscr{F}$ is given as follows:

$$\mathcal{L}_{FD}^* = \mathcal{L}_{FD}(\dot{\boldsymbol{F}}, \dot{\boldsymbol{F}}') = \frac{\sum_{j=1}^{N} d_j \cdot I_j}{\sum_j I_j} \tag{14}$$

We exclusively apply $\mathcal{L}_{FD}^*$ to labeled images, with a weight of $\lambda_{FD}$ set to 0.005, in order to prevent overfitting in the contexts of UDA and semi-supervised segmentation. However, we observe a performance drop in fully supervised segmentation when $\mathcal{L}_{FD}^*$ is applied. This suggests that the regularization might impede the exploitation of abundant supervision. Consequently, we set $\lambda_{FD}$ to 0 in the fully supervised setting.

## A.4 TRAINING AND INFERENCE DETAILS

### A.4.1 TRAINING

• **Augmentation**: PiXL employs several image augmentation techniques, including random cropping and random flipping. Initially, images from GTA5, SYNTHIA, and Cityscapes datasets are resized to 2560×1440, 2560×1520, and 2048×1024, respectively. Subsequently, 1024×1024 patches are randomly cropped from each image, with random flips applied as well. Unlike previous methods that downsampled the original images, this augmentation strategy preserves more image details at the cost of contextual information, intensifying the bias of local distribution $g$. However, PiXL excels in handling those issues.

• **Crop in Multi-Resolution Feature Extraction**: In the Multi-Resolution Feature Extraction module, an augmented 1024×1024 image is simultaneously cropped into a 512×512 high-resolution patch $X^H$ and resized into a 512×512 low-resolution patch $X^L$. During the initial 12000 iterations, random cropping is employed. Afterward, regions with high entropy are selectively cropped, aiding PiXL in focusing on the drift pixels.

• **ClassMix**: We have integrated ClassMix (Olsson et al., 2021) into PiXL to enhance data augmentation. Specifically, for two images $X_i$ and $X_j$, we randomly select half of the classes from $X_i$, mix them with $X_j$, and produce an augmented image $X_j'$. Following Tranheden et al. (2021), we apply color jitter and Gaussian blur to $X_j'$. Subsequently, both $X_i$ and the augmented image $X_j'$ are input into PiXL, as depicted in Fig. 5.

• **Pseudo label generation**: An EMA-updated version of PiXL is employed as the teacher model to generate pseudo labels for unlabeled images in UDA and semi-supervised segmentation. We denote the parameters in the teacher model and the back-propagated PiXL at training step $t$ as $\boldsymbol{W}_{EMA}^{(t)}$ and $\boldsymbol{W}_{PiXL}^{(t)}$, both initialized to be the same. The EMA update process is executed in accordance with the formula presented in equation 16.

$$\boldsymbol{W}_{EMA}^{(t)} = \begin{cases} \boldsymbol{W}_{PiXL}^{(t)}, t = 0 \\ \lambda_{EMA}^{(t)} \cdot \boldsymbol{W}_{EMA}^{(t-1)} + (1 - \lambda_{EMA}) \cdot \boldsymbol{W}_{PiXL}^{(t-1)}, t \geq 1 \end{cases} \tag{15}$$

The $\lambda_{EMA}^{(t)}$ at step $t$ is illustrated as follows:

$$\lambda_{EMA}^{(t)} = \min(1 - \frac{1}{t+1},\ 0.999) \tag{16}$$

• **Entropy** In PiXL, we employ Information Entropy (entropy) as the measurement criteria to split the pixels into joint and drift pixels. For a pixel feature representation $\boldsymbol{f}$ with its predicted probability distribution, its entropy $h$ is given by:

$$h = -\sum_{k=1}^{C} p_k \log p_k. \tag{17}$$

where $C$ is the number of classes and $p_k$ is the probability of the pixel belonging to class $k$.

### A.4.2 INFERENCE

We employ a sliding window strategy during the inference stage to make predictions, similar to the baseline HRDA (Hoyer et al., 2022b). The sliding window has a size of 1024×1024, with an overlap of $512 \times 512$. Each patch within the sliding window is predicted by combining results from the $512 \times 512$ high-resolution and low-resolution inputs.

## A.5 QUALITATIVE ANALYSIS

This section qualitatively analyzes segmentation results in semi-supervised and fully-supervised scenarios to highlight PiXL's effectiveness and robustness. As shown in Fig. 7, our results maintain precision and consistency across major classes like *road*, *sidewalk*, and *car*, as well as minor classes such as *pole*, *bicycle*, and *traffic sign*, even when the number of labeled images decreases significantly. This further validates the efficacy of pixel learning, particularly in label-scarce scenarios.

We also examine several failure cases of our PiXL in Fig. 8. Pixels located at boundaries pose challenges for accurate recognition. For instance, distinguishing pixels at the boundaries of categories like *fence* and *traffic sign*, or *terrain* and *road*, can be error-prone. Additionally, distinguishing pixels between categories with similar semantic information is also challenging, especially *sidewalk* and *road*, as depicted in Fig. 8. Finally, some minor categories with very few pixels in a region can be overlooked, such as *pole*.

## A.6 FUTURE WORK

Through extensive experimentation, we have substantiated the efficacy of the pixel learning scheme. This approach delves the segmentation model into the pixel level, significantly emphasizing the tailored learning process for each pixel. The proposed pixel learning scheme inherently aligns with the per-pixel recognition attributes integral to semantic segmentation. It also offers a unified framework for diverse semantic segmentation scenarios by accounting for pixel-level variances. To further harness the advantages of pixel learning, the following considerations merit attention:

• Pixel learning serves as a plug-and-play framework that seamlessly integrates with diverse backbones and decoders, enhancing the performance of existing models.

• While multi-resolution strategies are common in semantic segmentation, they are optional in our scheme. In fully supervised segmentation tasks, our approach, which incorporates a multi-resolution strategy, conserves space compared with existing models. However, further optimization opportunities exist.

• While the simple choice of $\eta$ demonstrates the model's robustness compared with the baseline, it's important to consider varying pixel-level variances across tasks and differences in bias between local and global distributions. Further research on the selection strategy for $\eta$ or alternative pixel feature partitioning methods is valuable.

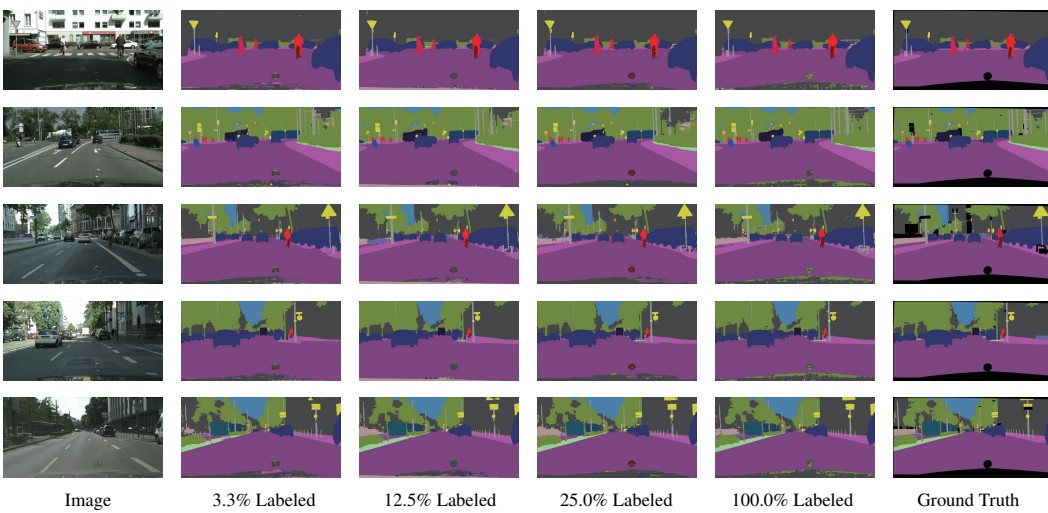

|        | Image | 3.3% Labeled | 12.5% Labeled | 25.0% Labeled | 100.0% Labeled | Ground Truth |

Figure 7: More qualitative examples on PiXL (HRDA) on different ratios of labeled images. The segmentation results are precise and consistent in both major categories and minor categories.

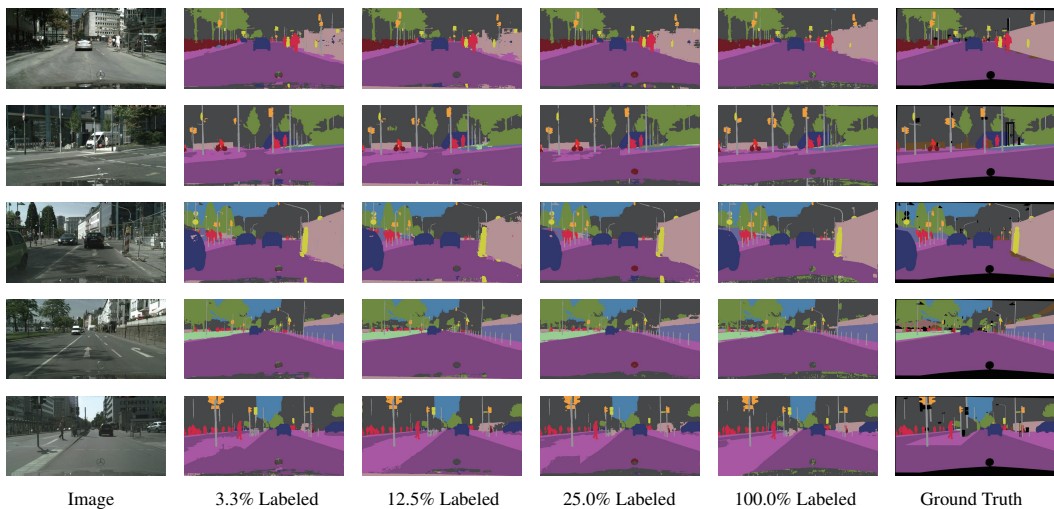

Figure 8: Failure cases of PiXL (HRDA) in the semi-supervised and fully-supervised setting. The ability to recognize similar pixels lying in boundaries requires further improvement in categories like *sidewalk*, *road*, *terrain*, and *fence*.

