# OpenReview forum: "Diving Segmentation Model into Pixels"
_ICLR.cc/2024/Conference — ICLR 2024 poster_

### Official Review · Reviewer_GSD6 · 2023-10-30

**Soundness:** 3 good
**Presentation:** 3 good
**Contribution:** 3 good
**Rating:** 6
**Confidence:** 3

**Summary:**

This paper proposed a  pixel learning framework for semantic segmentation. Intra-image, inter-image, inter-domain pixels variances are considered in this framework. The framework is elaborate, which consists of four components, i.e., Multiple Resolution Feature Extraction, Pixel Level Sub-Domain Partition, Adaptive Prototype Generation, and Drift Pixels Alignment. The motivation of this paper is interesting. The experimental results demonstrate the effectiveness of the proposed method.

**Strengths:**

(1) Pixel variance is important in semantic segmentation. This paper proposed a new solution.

(2) This framework is flexible, it is quite easily to perform different semantic segmentation tasks.

(3) The performance is good.

**Weaknesses:**

(1) The authors did not report the results on higher resolution images, is that because too many pixels should be considered?

(2) In semantic segmentation, contextual information is quite important to assign a class label to a pixel.  But this paper discards the context in some extent. Is this reasonable?

(3) Pixel-level contrastive learning is widely used in unsupervised semantic segmentation, both local and global relations are considered. In these methods, global pixel features are usually store in a memory bank.  The differences with these method should be given in detail.

(4) In Table 1, the proposed method performs worse with 1/8 and 1/4 than 1/30. The authors should explain this.

(5) In addition of intra-image, inter-image pixel relations, this method also considers the inter-domain one, but this is not presented in abstract.

**Questions:**

see the weaknesses

---

> ### Author Response · Authors · 2023-11-21
> **To Reviewer GSD6 (1/5)**
>
> We would like to express our sincere gratitude for your valuable comments and suggestions. We will respond to your concerns one by one.
>
> ## W1:
> The authors did not report the results on higher resolution images, is that because too many pixels should be considered?
>
> ## R1:
> We sincerely appreciate the insightful perspectives and valuable suggestions provided by the reviewer. They have highlighted the computational cost and efficiency challenges associated with the pixel learning scheme.
>
> Indeed, considering all pixels in high-resolution images does impose pressure on memory and computational efficiency. These feedbacks underscore a compelling direction for further exploration of the pixel learning scheme.
>
> Although our PiXL is trained on images with moderate resolution, its performance is competitive, even when compared with models trained on higher-resolution images in semi-/fully-supervised settings. **Moreover, experiments on tough UDA and label-scarce semi-supervised settings indicate the merits of our pixel learning scheme when confronted with complex real-world scenarios.**
>
> Regarding efficiency, authors in [1] also encountered such an issue. They proposed calculating the loss on some sampled points instead of the whole mask. This could inspire us to delve into the pixel sampling mechanism to overcome the efficiency burden in pixel learning.
>
> Diverging from the emphasis of this work on validating the effectiveness of the pixel learning scheme, we intend to delve deeper into this efficiency issue in a new study in the future.
>
> We hope our responses address your concerns.
>
> [1] Cheng, Bowen, et al. "Masked-attention mask transformer for universal image segmentation." *Proceedings of the IEEE/CVF conference on computer vision and pattern recognition*. 2022.
>
> ## W2:
> In semantic segmentation, contextual information is quite important to assign a class label to a pixel. But this paper discards the context in some extent. Is this reasonable?
>
> ## R2:
> Thank you for your valuable insights. We highly appreciate the significance of contextual information in semantic segmentation.
>
> In this work, we have devoted concerted efforts to incorporate contextual information into pixel feature representation.
>
> Firstly, the backbone in PiXL extracts multi-scale feature representations and adopts the atrous spatial pyramid pooling (ASPP) $^{[1]}$ module to fuse the extracted multi-scale features to embed more contextual information into pixel features.
>
> Secondly, considering that different categories have varying dependencies on low-resolution input, which provides more abundant contextual information, we propose the adaptive prototype generation module to enhance further the context information for classes like *sky*, *road,* *building*, etc.
>
> These designs succeed in encompassing contextual information into pixel features and are conducive to PiXL's competitive performance under various settings.
>
> Moreover, exploring the design of a purely pixel-based feature extraction method and integrating contextual information within it represents an interesting issue within pixel learning, which will be delved deeper in future endeavors.
>
> We hope our responses address your concerns.
>
> [1] Chen, Liang-Chieh, et al. "Deeplab: Semantic image segmentation with deep convolutional nets, atrous convolution, and fully connected crfs." *IEEE transactions on pattern analysis and machine intelligence* 40.4 (2017): 834-848.

---

> ### Author Response · Authors · 2023-11-21
> **To Reviewer GSD6 (2/5)**
>
> ## W3:
> Pixel-level contrastive learning is widely used in unsupervised semantic segmentation, both local and global relations are considered. In these methods, global pixel features are usually store in a memory bank. The differences with these method should be given in detail.
> ## R3:
>
> Thank you for your insightful comments on our approach. We will systematically explain the distinctions and advantages of our method compared to other pixel-level contrastive learning methods, like storing global information in a memory bank.
>
> Our PiXL differs from the existing methods, employing pixel-level contrastive learning.
>
> Firstly, contrary to the existing methods $^{[1][2][3]}$, which pursue consistency feature representation categorically by utilizing both local and global relations, our PiXL concentrates on coping with local relations. Furthermore, even from the perspective of exploring local information, our approach differs from some existing methods. We emphasize the difference among pixels and advocate designing tailored learning strategies for different pixels, considering pixel-level variance and that the trivial employment of pixel-level contrastive learning may enlarge misleadings from some pixels with poor feature representation. Specifically, **we formulate each image as a local distribution of pixels and hypothesize that a few pixels within the local distribution conform to the categorical implicit global distribution while the others do not, denoted as joint pixels and drift pixels.** **Thus, our PiXL continuously aligns drift pixels from two local distributions to the local prototypes generated from the joint pixels, hoping to address the intra-/inter-image and inter-domain variance ceaselessly.**
>
> Secondly, current methods, such as employing a memory bank to store global pixels or prototypes $^{[1][2][3]}$, may not be optimal. Considering the pixel-level variance, as we summarized in Figure 1, the global pixel features from different images or different iterations stored in the memory bank may also vary greatly, generating misleading signals that confuse anchor pixel features and hamper the alignment. Moreover, the methods ^{[3]} maintaining a single or storing multiple continually updated global prototypes in the memory bank to represent the implicit global distribution of each class also fail to achieve optimal because the complexity of the global distribution exceeds the representation capacity of prototypes.
>
> Compared with these methods, PiXL achieves competitive performance by only concentrating on mining the local relations thanks to two merits of our model: (1) The local prototypes generated from joint pixel features suppress misleading signals originating from pixels with subpar feature representation. (2) Using local prototypes solely for aligning drifting pixels within two local distributions exploits their representational power while avoiding exceeding the limited capacity of prototypes
>
> These experiments on different settings validate the effectiveness of our proposed pixel learning scheme, verifying the merits of diving the segmentation models into pixel level and addressing pixel-level variance, which is a promising research field.
>
> Undoubtedly, integrating global information into the pixel learning scheme is beneficial. However, existing methods to mine and exploit global relationships may require further optimization. This aspect remains a worthwhile pursuit in the field of pixel learning. In our future work, we will concentrate on how to effectively combine local and global relationships under the context of pixel learning and design novel models.

---

> ### Author Response · Authors · 2023-11-21
> **To Reviewer GSD6 (3/5)**
>
> ## R3 (Cont.)
> The comparison between PiXL and other pixel-level contrastive learning methods is summarized here.
>
> In fully supervised semantic segmentation:
>
> | Method | Image Size      | Cityscapes |
> | ------ | --------------- | ---------- |
> | [1]    | 512$\times$1024 | 82.40      |
> | PiXL   | 512$\times$512  | 82.74      |
>
> In semi-supervised semantic segmentation:
>
> | Method           | Image Size      | Cityscapes(1/30) | Cityscapes(1/8) | Cityscapes(1/4) |
> | ---------------- | --------------- | ---------------- | --------------- | --------------- |
> | [2]              | 512$\times$1024 | 64.90            | 70.10           | 71.70           |
> | PiXL (DeepLabV3) | 512$\times$512  | 70.62            | 75.20           | 78.20           |
> | PiXL (HRDA)      | 512$\times$512  | 71.73            | 76.37           | 78.91           |
>
> In unsupervised domain adaptation:
>
> | Method | Road | S.Walk | Build. | Wall | Fence | Pole | Tr.Light | Tr.Sign | Veget. | Terrain | Sky  | Person | Rider | Car  | Truck | Bus  | Train | M.Bike | Bike | mIoU |
> | ------ | ---- | ------ | ------ | ---- | ----- | ---- | -------- | ------- | ------ | ------- | ---- | ------ | ----- | ---- | ----- | ---- | ----- | ------ | ---- | ---- |
> | [3]    | 96.3 | 73.6   | 89.6   | 53.7 | 47.8  | 53.8 | 60.8     | 60.0    | 89.9   | 48.8    | 91.5 | 74.6   | 45.1  | 93.1 | 74.8  | 73.8 | 51.5  | 60.3   | 65.3 | 68.7 |
> | PiXL   | 97.0 | 77.6   | 91.1   | 59.9 | 54.1  | 57.2 | 64.8     | 69.1    | 91.5   | 51.8    | 94.8 | 80.5   | 57.3  | 94.6 | 83.8  | 88.7 | 78.0  | 65.6   | 67.8 | 75.0 |
>
> We hope our responses address your concerns.
>
> [1] Wang, Wenguan, et al. "Exploring cross-image pixel contrast for semantic segmentation." *Proceedings of the IEEE/CVF International Conference on Computer Vision*. 2021.
>
> [2] Alonso, Inigo, et al. "Semi-supervised semantic segmentation with pixel-level contrastive learning from a class-wise memory bank." *Proceedings of the IEEE/CVF International Conference on Computer Vision*. 2021.
>
> [3] Xie, Binhui, et al. "Sepico: Semantic-guided pixel contrast for domain adaptive semantic segmentation." *IEEE Transactions on Pattern Analysis and Machine Intelligence* (2023).

---

> ### Author Response · Authors · 2023-11-21
> **To Reviewer GSD6 (4/5)**
>
> ## W4:
> In Table 1, the proposed method performs worse with 1/8 and 1/4 than 1/30. The authors should explain this.
>
> ## R4:
>
> Thanks for your thoughtful comments and valuable feedback!
>
> In the semi-supervised setting, our PiXL demonstrates competitive performance, especially when only 3.3% of images have annotations, indicating that the pixel learning scheme excels in coping with complex, real-world scenarios. In 1/4 and 1/8 settings, PiXL surpasses all methods trained on images with the same resolution and even outperforms some methods trained on higher-resolution images. Even as the number of annotated images decreases, our model exhibits remarkable robustness.
>
> In addition, our primary focus in this study is to explore a new semantic segmentation scheme and develop an adaptable, efficient model capable of handling diverse semantic segmentation settings, distinct from models designed solely for specific scenarios. Comprehensive qualitative, quantitative, and visual analyses have validated the success of our initial exploration. In the subsequent new studies on the pixel learning scheme, we aim to further optimize and enhance its capabilities.
>
> We hope our responses address your concerns.

---

> ### Author Response · Authors · 2023-11-21
> **To Reviewer GSD6 (5/5)**
>
> ## W5:
> In addition of intra-image, inter-image pixel relations, this method also considers the inter-domain one, but this is not presented in abstract.
>
> ## R5:
> Thank you for your valuable feedback. We have diligently revised the manuscript.
>
> Moreover, as depicted in Figure 1, the intra-/inter-image, inter-domain, and label-error-related variance collectively contribute to significant heterogeneity among pixel features belonging to the same category, leading to a more dispersed distribution in the feature space.
>
> Correspondingly, the proposed pixel learning aims to alleviate pixel-level variance, meaning to enhance the consistency among pixel features belonging to the same category and seek a more compact distribution in the feature space.
>
> The revised version of the abstract is given as follows:
>
> > “More distinguishable and consistent pixel features for each category will benefit the semantic segmentation under various settings. Existing efforts to mine better pixel-level features attempt to explicitly model the categorical distribution, which fails to achieve optimal due to the significant pixel feature variance. Moreover, prior research endeavors have scarcely delved into the thorough analysis and meticulous handling of pixel-level variances, leaving semantic segmentation at a coarse granularity. In this work, we analyze the causes of pixel-level variance and raise the concept of **pixel learning** to concentrate on the tailored learning process of pixels, handle pixel-level variance, and enhance the segmentation model's per-pixel recognition capability. **Under the context of the pixel learning scheme, each image is viewed as a distribution of pixels, and pixel learning aims to pursue consistent pixel representation inside an image, continuously align pixels from different images (distributions), and eventually achieve consistent pixel representation for each category, even cross-domains. We proposed a pure pixel-level learning framework, namely PiXL, which consists of a pixel partition module to divide pixels into sub-domains, a prototype generation, a selection module to prepare targets for subsequent alignment, and a pixel alignment module to guarantee pixel feature consistency intra-/inter-images, and inter-domains**. Extensive evaluations of multiple learning paradigms, including unsupervised domain adaptation and semi-/fully-supervised segmentation, show that PiXL outperforms state-of-the-art performances, especially when annotated images are scarce. Visualization of the embedding space further demonstrates that pixel learning attains a superior representation of pixel features. The code will be available upon acceptance.”
>
> We hope our responses address your concerns.

---

### Official Review · Reviewer_YSzS · 2023-10-31

**Soundness:** 3 good
**Presentation:** 3 good
**Contribution:** 2 fair
**Rating:** 6
**Confidence:** 3

**Summary:**

In this paper, the authors introduce the pixel learning scheme by treating each image as a local distribution of pixels. The PiXL framework, which segregates pixels within a given local distribution into sub-domains: joint pixels and drift pixels, is proposed. Then, the PiXL employs an asymmetric alignment approach to align drift pixels with the joint pixels, effectively addressing pixel-level variance in a
divide-and-conquer manner. Extensive experiments confirm PiXL’s performance, especially demonstrating promising results in
label-scarce settings.

**Strengths:**

This paper  proposes a novel pixel learning scheme to dive semantic segmentation models into the pixel level by treating an image as a distribution of pixels. This advocates addressing pixel-level variance to enhance the segmentation model’s per-pixel recognition capability.
The strengths are as follows:
1. This paper proposed PiXL, a pure pixel-level learning framework that executes pixel-level intra- and inter-distribution (image) alignment within the context of pixel learning.
2. Extensive quantitative and qualitative experiments in various settings of semantic segmentation confirm the effectiveness of PiXL, demonstrating the feasibility and superiority of the pixel learning scheme, which deserves further exploration.
3. The writing is clear and well-reading.

**Weaknesses:**

The weakness are as follows:
1. Some description is not very clear. For example, in equation 4, "PiXL determines the threshold ...", how to determine the threshold is not presented. why " that pixel partitioning is performed separately...considering the entropy gap across images."?
2. "PiXL employs entropy as the criteria to segregate the pixel features in g into joint pixels and drift
pixels." how to compute the entropy?
3. the paper validate the effectiveness on HRDA model, but if the proposed methods can be applied to general semantic segmentation methods is not verified.
4. In table 3, the proposed method cannot show state-of-the-art performance compared with other methods. The authors should prove its effectiveness.

**Questions:**

The questions are summarised with weakness part.

---

> ### Author Response · Authors · 2023-11-21
> **To by Reviewer YSzS**
>
> Thank you for your valuable feedback! We have carefully revised the manuscript according to your suggestions and provided more details, especially about the entropy calculation. Moreover, we will explain your concerns point by point.
>
> ## W1:
> Some description is not very clear. For example, in equation 4, "PiXL determines the threshold ...", how to determine the threshold is not presented. why " that pixel partitioning is performed separately...considering the entropy gap across images."?
>
> ## W2:
> "PiXL employs entropy as the criteria to segregate the pixel features in g into joint pixels and drift pixels." how to compute the entropy?
>
> ## R1 & R2:
> 1. We appreciate your thoughtful and constructive feedback and provide more details on the calculation of entropy in the Appendix as follows:
>
>    > ''In PiXL, we employ Information Entropy (entropy) as the measurement criteria to split the pixels into joint and drift pixels. For a pixel feature representation $\boldsymbol{f}$ with its predicted probability distribution, its entropy h is given by:
>    > $$
>    > \begin{equation}
>    > \begin{aligned}
>    > &h = - \sum_{k = 1}^{C}p_{k}\log p_{k}.
>    > \end{aligned}
>    > \end{equation}
>    > $$
>    > where $C$ is the number of classes and $p_{k}$ is the probability of the pixel belonging to class $k$."
>
>    The larger $h$ of a pixel means the greater uncertainty of its prediction. The entropy of a one-hot distribution is $0$, indicating no uncertainty. Thus, the pixels with smaller entropy imply less uncertainty and are referred to as joint pixels, which are assumed to conform to the implicit global distribution, while the pixels with larger entropy are denoted as drift pixels.
>
> 2. The threshold $\epsilon$ is determined by the sorted pixel entropy from each image. Specifically, given the extracted pixel features $F^L$ and $F^H$ from low-resolution input $X^L$ and high-resolution crop $X^H$, respectively, we compute their corresponding pixel entropy $h$ for each pixel feature and sort them according to their entropy. The pixel entropy at the $1 - \eta$ quantile is selected as the threshold $\epsilon$. The details are provided in Section 3.4 as depicted in Equation (4) and c. part in Figure 2.
>
> 3. Considering the pixel-level variance summarized in Figure 1, the range of pixel entropy may differ between two images. Given the most extreme condition where the entropy of pixels in image $\textit{A}$ are all larger than the ones in image $\textit{B}$, conducting partition on the pixels from two images instead of on each image separately will cause the failure to divide the pixels within each distribution into sub-domains and eventually hinders the intra-distribution alignment.
>
> We hope our responses address your concerns.
>
> ## W3:
> the paper validate the effectiveness on HRDA model, but if the proposed methods can be applied to general semantic segmentation methods is not verified.
>
> ## R3:
> Although our experiments are primarily based on HRDA model, our method is plug-and-play and applicable to other semantic segmentation models. In semi-supervised segmentation, apart from the PiXL based on HRDA, we also implemented a DeepLabV3+-based PiXL. Their competitive performance indicates that our methods can be easily applied to many existing semantic segmentation models. Please refer to the Network architecture paragraph in Section 4.3, Table 1, and Section A.2 in the appendix for more details.
>
> We hope our responses address your concerns.
>
> ## W4:
> In table 3, the proposed method cannot show state-of-the-art performance compared with other methods. The authors should prove its effectiveness.
>
> ## R4:
> Thanks for your thoughtful comments and valuable feedback!
>
> We elaborate on your concerns about the performance of PiXL in the response to all reviewers. Please refer to that for details.
>
> Moreover, we have summarized the key points here to address your concerns better.
>
> In this study, our primary focus is exploring a novel semantic segmentation scheme and devising a versatile and efficient model that caters to diverse settings in semantic segmentation. Our approach differs from proposing a model specifically tailored, developed, or optimized for fully supervised settings.
>
> Despite this, as a first exploration of the pixel learning scheme, our PiXL model has showcased commendable performance, particularly in demanding unsupervised domain adaptation and semi-supervised settings where labeled data is limited. This underscores the suitability of the pixel learning scheme to address the requirements of intricate real-world scenarios.
>
> Our future research endeavors aim to further optimize the pixel learning scheme by strengthening global distribution constraints, integrating contextual information, emphasizing model efficiency and lightweight designs, and further bolstering the model's performance across diverse task settings.
>
> We hope our responses address your concerns.

---

### Official Review · Reviewer_nhEN · 2023-10-31

**Soundness:** 3 good
**Presentation:** 3 good
**Contribution:** 3 good
**Rating:** 6
**Confidence:** 3

**Summary:**

This paper takes pixel-level distribution (local distribution) into consideration, and proposed Pixel Level Sub-Domain Partition Module (PSP), Adaptive Prototype Generation Module (APG); Drift Pixels Alignment Module(DPA) modules, the effectiveness of which is proved in ablation study.
First, the PSP module divides all features (with multiscale feature extraction) into Joint pixel features and Drift Pixel features based on the entropy of each segmentation pixels corresponding to each pixel feature. For Joint ones, they use APG to generate local feature prototypes based on their semantic classes, while the Drift ones would be pulled by the prototypes extracted from APG using info NCE loss.
The prototypes from APG is the mean value of pixel features which belongs to Joint pixel features in two samples. From which, the paper argues that can get intra-image and inner-image information.
Finally, the effectiveness of these module is proved in unsupervised domain adaptation, semi-supervised semantic segmentation together with fully-supervised semantic segmentation.

**Strengths:**

1. This paper is well-written and nicely organized.
2. Extensive experiments have been conducted and a number of quantitative and qualitative results are shown, demonstrating the effectiveness of the proposed method.

**Weaknesses:**

1. The method seems incremental. The paper generates class prototype from two image features and push or pull image features based on these prototypes. Their novelty lies in the partition of joint features and drifted features, the selection of high resolution feature prototypes and low resolution prototypes, which seems to be triky.

**Questions:**

see weakness

---

> ### Author Response · Authors · 2023-11-21
> **To Reviewer nhEN**
>
> Thank you for your valuable comments on our manuscript. We will address your questions regarding our work one by one.
>
> ##  W1:
> The method seems incremental.
>
> ## R1:
>
> The PiXL is an end-to-end semantic segmentation model, which is not a two-stage method or based on a well-trained model. In PiXL, the backbone is initialized by ImageNet pre-trained parameters, while the decoder is randomly initialized following the conventional practice. Then, the pixel features are continuously furnished to pursue an intra-class compact and inter-class separable pixel feature distribution. Figure 4 validates that the pixel feature distribution produced by PiXL is better than the SOTA method.
>
> ## W2:
> The selection of high resolution feature prototypes and low resolution prototypes seems to be tricky.
>
> ## R2:
>
> We propose this selection mechanism in Adaptive Prototype Generation module based on the consideration that different categories have varying dependencies on low-resolution input $X^L$ and high-resolution crop $X^H$. A higher resolution crop $X^H$ can benefit the recognition of smaller targets, like person and pole, as it provides abundant details, while a larger field of view in low-resolution input $X^L$ aids in better context perception, thereby improving the segmentation of larger objects, such as sky and road. Thus, this selection mechanism aims to adaptively embed context or details into pixel features of different categories. The ablation study in Table 4 verifies its effectiveness. It's worth noting that this selection mechanism is not handcrafted but adaptively applied according to the pixels in low-resolution input $X^L$ and high-resolution crop $X^H$ as depicted in d. part of Figure 2 and Equation (8).

---

### Official Review · Reviewer_L3L6 · 2023-11-04

**Soundness:** 3 good
**Presentation:** 3 good
**Contribution:** 3 good
**Rating:** 6
**Confidence:** 3

**Summary:**

The paper proposes  a pixel-level learning framework, PiXL, for semantic segmentation. The framework consists of a pixel partition module,  a prototype generation and selection module, and a pixel alignment module. The pixel partition module separate pixels features into joint and   drift pixels based on their entropy. The prototype generation module is to select the most meaningful pixels. The pixel aligment module adopts contrastive learning to align pixel features intra and inter-distribution. The effectiveness of proposed framework and components are experimentally validated on three public datasets: GTA5, SYNTHIA, and Cityscapes.

**Strengths:**

1. The idea of investigating semantic segmentation from pixel feature distribution perspective is novel.
2. The proposed pixel learning framework and its components are technically solid and innovative.
3. The motivation and the underlying principles of designing the framework and each components are clearly presented and explained, so that it is easy to follow the work.
4. The experiments are extensive and solid.

**Weaknesses:**

The experiment results of the proposed results are not much better than previous works.

**Questions:**

no

---

> ### Author Response · Authors · 2023-11-21
> **To Reviewer L3L6**
>
> ## W1:
> The experiment results of the proposed results are not much better than previous works.
>
> ## R1:
> Thanks for your insightful comments and valuable feedback!
>
> We elaborate on your concerns about the performance of PiXL in the response to all reviewers. Please refer to that for details.
>
> Moreover, we have summarized the key points here to address your concerns better.
>
> This work's primary concern is exploring a novel semantic segmentation scheme to provide a novel perspective on image comprehension. Specifically, we propose the **Pixel Learning** scheme by formulating each image as a distribution of pixels, addressing the pixel-level variance to recognize each pixel accurately.
>
> Given this, the meticulous optimization of backbones or task settings is not the primary focus of this work. Nevertheless, our PiXL performs competitively across various settings, affirming the effectiveness of the pixel learning scheme and suggesting its potential for further exploration.
>
> In particular, PiXL excels in challenging scenarios like unsupervised domain adaptation and semi-supervised settings with very limited labeled data, verifying PiXL's effectiveness and robustness in complex real-world scenarios. Qualitative analysis and visualization have further illustrated the promising pixel learning scheme.
>
> In the future, we will delve deeper and further optimize the pixel learning scheme from global constraint enhancement, contextual information integration, model efficiency, and lightweight, respectively.
>
> We hope our responses address your concerns.

---

> > ### Comment · Reviewer_L3L6 · 2023-12-01
> >
> > Thanks for your response.

---

### Author Response · Authors · 2023-11-21
**To all reviewers**

Many thanks for your positive comments and insightful questions about our paper!

We carefully considered these suggestions and conducted comprehensive revisions to the manuscript. In the submitted revised manuscript, revision are marked as red.

**Moreover, we would like to elaborate on your concerns regarding the model's performance, including:**

- **Reviewer L3L6 Q1**
- **Reviewer YSzS Q4**
- **Reviewer GSD6 Q3**

In this work, we aim to explore a novel semantic segmentation scheme called **Pixel Learning** by **formulating each image as a distribution of pixels, addressing the pixel-level variance to recognize each pixel accurately**. Thus, our main concern in this work is to design a universal and effective model under the context of pixel learning that caters to diverse settings in semantic segmentation. This endeavor aims to contribute to **a novel perspective on image comprehension** in future computer vision research, proposing **a more fine-grained learning scheme**.

Comparatively, tailored optimization of the model for specific task settings or backbones, such as fully supervised scenarios, though it could yield superior outcomes, is not the primary focus of this work. Nevertheless, as an initial exploration of pixel learning, our experiments and analysis have amply demonstrated the effectiveness of the pixel learning scheme and its potential for further investigation.

The fully supervised, semi-supervised, and unsupervised domain adaptation experiments showcase PiXL's effectiveness and robustness in complex real-world scenarios. Notably, it excels in challenging scenarios like unsupervised domain adaptation and semi-supervised settings with very limited labeled data, attesting to its adaptability and label-efficient property. Specifically, PiXL outperforms the SOTA methods **in the most difficult UDA setting**. PiXL is significantly superior **in semi-supervised settings when only 3.3% of images are labeled**. In 1/8 and 1/4 images labeled and fully supervised settings, PiXL outperforms all methods trained on images with the same resolution, surpassing some approaches trained on higher-resolution images.

These experiment results, qualitative analysis(Figure 6), and pixel feature visualization(Figure 4) have effectively validated the success of the exploration of the novel semantic segmentation scheme, indicating that diving into pixel level to analyze and address the pixel-level variance by tailoring learning strategy for different pixels will produce more distinguishable and consistent pixel features, thus facilitate semantic segmentation model to recognize each pixel more precisely.

Nevertheless, as the first exploration of pixel learning, we primarily focus on investigating pixel-level variations within local distributions. Specifically, we concentrate on continuously aligning drift pixels with the joint ones across two local distributions to finally pursue a better global distribution categorically. Despite the commendable performance shown by solely conducting pixel learning from a local perspective, there remain several other perspectives worth exploring to enhance the model's performance further in pixel learning scheme, including explicitly constraining the consistency of pixel features in global distributions, integrating more contextual information into pixel features, designing more efficient and lightweight models to cater to efficiently learning on higher-resolution images, etc.

For example, the work [1],  despite lagging behind PiXL in a fully supervised setting, demonstrated that adopting global constraints, such as global pixels or prototypes, could further enhance the segmentation performance of the model. This aspect is worthy of further exploration in future research on pixel learning.

| **Method** | **Image Size**  | **Cityscapes** |
| ---------- | --------------- | -------------- |
| [1]        | 512$\times$1024 | 82.40          |
| PiXL       | 512$\times$512  | 82.74          |

In our forthcoming research endeavors, we aim to explore global constraints, contextual information, and strategies for model efficiency and lightweight design to optimize the pixel learning scheme further and offer a more universal and fine-grained semantic segmentation framework.

We hope our responses address your concerns.

[1] Wang, Wenguan, et al. "Exploring cross-image pixel contrast for semantic segmentation." *Proceedings of the IEEE/CVF International Conference on Computer Vision*. 2021.

---

### Meta-Review · Area_Chair_UJcW · 2023-12-12

**Metareview:**

This paper proposes a pixel-wise feature learning method by viewing an image as a local distribution of pixels and the goal of learning is to create a consistent pixel representation by semantic categories within and across images.  It consists of pixel partitioning, prototype generation, pixel selection and alignment.  Benchmarked on GTA5, SYNTHIA, and Cityscapes, experimental gains are observed on unsupervised domain adaptation and semi-/fully-supervised segmentation, especially when there are limited labeled data.

The strengths of the paper are clear writing, flexible deployment on diverse tasks, and extensive experimentation.  The weaknesses of the paper are incremental novelty, limited gains, and high computational costs for large images.

The paper has received 4 reviews, with a consensus of rating 6.  The authors have provided further clarification and additional experimental results.  The statistical distribution perspective is novel and valuable in motivating the model development.  Weighing all the pros and cons, the AC recommends acceptance.

The paper misses several important pixel-wise feature learning papers for segmentation, e.g.,
[1] SegSort: Segmentation by Discriminative Sorting of Segments, ICCV 2019
[2] Self-supervised Visual Representation Learning from Hierarchical Grouping, NeurIPS 2020
[3] Universal Weakly Supervised Segmentation by Pixel-to-Segment Contrastive Learning, ICLR 2021
[4] Unsupervised Hierarchical Semantic Segmentation with Multiview Cosegmentation and Clustering Transformers, CVPR 2022

**Justification For Why Not Higher Score:**

Incremental technical novelty and limited empirical gains.
Reviewers' lukewarm reception of the paper.

**Justification For Why Not Lower Score:**

Novel perspective on pixel-wise feature learning to align the local distribution from a single image consistently with the global distribution from all the images.

---

### Decision · Program_Chairs · 2024-01-16

Accept (poster)